# Designing ethical souvenirs to sustain the cultural integrity of Dunhuang heritage

**Qiuxia Zhu** [1,2]*, **Rizal Rahman**[2]

**1** Faculty of Art, Lanzhou University of Finance and Economics, Lanzhou, China, **2** Faculty of Design and Architecture, Universiti Putra Malaysia, Seri Kembangan, Malaysia

* qxzhu826413@gmail.com

## Abstract

Mass-produced tourism souvenirs often dilute the cultural integrity of heritage sites, yet the mechanisms by which design philosophy and ethics sustain authenticity remain underexplored. This study investigates how designers' philosophical stances and ethical practices shape authentic Dunhuang souvenirs. Adopting a qualitative single-case design, the investigation used purposive sampling to recruit 11 participants (eight designers and three cultural experts). Data from in-depth semi-structured interviews, non-participant observations, and document review were analysed via reflexive thematic analysis conducted concurrently with data collection. Three interlinked themes emerged: (1) design principles and philosophy, (2) ethical considerations, and (3) design implementation, coalescing into the Ethical Design for Cultural Integrity (EDCI) framework. The EDCI explains how authenticity and market relevance can be co-achieved by coupling value-laden design reasoning with procedural safeguards in production. Practical guidance includes: pre-brief checklists identifying non-negotiable cultural elements; embedded ethics procedures across decision points; and feedback-rich implementation cycles that preserve core motifs while meeting manufacturability and price envelopes. The study's originality lies in integrating philosophy, ethics, and implementation into a single process view of heritage product design and in operationalising ethics as procedural governance rather than declarative intent, offering a replicable pathway for culturally responsible souvenir development.

## 1. Introduction

Cultural integrity is a core principle of the UNESCO World Heritage Convention, which refers to keeping the completeness and authenticity of heritage sites so that their essential attributes remain intact for present and future generations [1]. This principle extends beyond physical conservation to encompass safeguarding intangible cultural values, ensuring that the beliefs, practices, and traditions of a community

**Data availability statement:** All relevant data are within the manuscript and its Supporting Information files.

**Funding:** The author(s) received no specific funding for this work.

**Competing interests:** The authors have declared that no competing interests exist.

are represented authentically [2,3]. In this study, cultural integrity refers to the preservation and authentic representation of a culture's beliefs, practices, and traditions [4], ensuring that heritage retains its relevance and importance over time.

Souvenir design serves as a tangible representation of a destination's identity and heritage and has long been a crucial aspect of cultural preservation and promotion [5,6]. When grounded in ethical principles, it can bridge tradition and modernity, creating innovative products that appeal to evolving consumer preferences while retaining authenticity [7,8]. However, the industry has faced mounting criticism for the proliferation of mass-produced, inauthentic, or culturally appropriated products that undermine cultural integrity [9,10]. Designers' roles thus extend far beyond aesthetics; they carry an ethical responsibility to uphold cultural values and traditions while navigating commercial pressures [11,12].

Past research has examined factors influencing the cultural integrity of souvenirs, such as the use of traditional materials and techniques, the involvement of local artisans, and the representation of cultural symbols and narratives [13–15]. Studies have also explored the economic and socio-cultural impacts of souvenirs, including their role in destination branding, local economic contribution, and tourism promotion [16,17]. The production and consumption of souvenirs often commodify material culture, transforming it into a transactional commodity between tourism providers and visitors [8,18]. While some designers adopt practical design principles and sustainable approaches to honor cultural traditions [19,20], most prior studies treat cultural preservation and design ethics as separate lines of inquiry [11,21], without exploring the mechanisms through which designers' moral principles and decision-making frameworks and cultural responsibilities influence the creation of authentic heritage products. There is a need for a comprehensive framework to evaluate the cultural integrity of souvenirs and to measure the effectiveness of different design strategies in real-world contexts.

This study addresses these gaps by examining the role of designer philosophy and ethics in sustaining cultural integrity within souvenir innovation. Using qualitative case analysis in Dunhuang, a UNESCO World Heritage site renowned for its Buddhist murals, this research proposes a Framework of Ethical Souvenir Design for Cultural Integrity. The framework synthesizes insights from in-depth interviews with designers and cultural experts to identify ethical principles, design values, and implementation strategies that reconcile cultural authenticity with market viability. The objectives of this study are to:

RO 1. Identify the fundamental ethical principles and values that guide designers in creating culturally authentic souvenirs.

RO 2. Assess the impact of designer philosophy and ethics of cultural integrity in souvenir design.

To achieve these objectives, the study will address the following research questions:

RQ 1. What are the fundamental ethical principles and values that guide designers in creating authentic souvenirs?

RQ 2. How do designer philosophy and ethics impact cultural integrity in souvenir design?

By addressing these questions, this study contributes to heritage design theory by integrating cultural integrity with design ethics, while providing actionable guidance for designers, policymakers, and industry stakeholders committed to ethical and sustainable souvenir development [22].

## 2. Literature review and theoretical background

### 2.1. Cultural integrity in heritage product design

Cultural integrity, defined as "a measure of the wholeness and intactness of the natural and/or cultural heritage and its attributes" [23] (p. 33). It is implicitly linked to cultural meaning, which includes historical, aesthetic, social, scientific, or spiritual values that make sites significant in the past, present, and future [24]. The conservation of heritage is also related to national identity and security as it ensures the continuation of a nation's cultural narrative [25].

In the context of heritage tourism, souvenirs operate as both economic commodities and cultural artefacts, serving as tangible carriers of meaning that connect visitors to local narratives and support community livelihoods [26,27]. However, the globalization of tourism has accelerated the spread of mass-produced, inauthentic, and culturally misappropriated products [28], threatening the integrity of cultural representation. Souvenirs' cultural integrity requires deliberate design strategies that reinforce originality, craftsmanship, and place-specific narratives [8,29]. Designers should also contend with evolving consumer preferences and competitive market demands, which can pressure them to compromise on cultural depth in favour of commercial appeal [30].

Authenticity is also closely linked to the protection of cultural heritage and providing visitors with authentic experiences. Designers should navigate evolving consumer preferences and competitive market demands, which may compel them to sacrifice cultural depth for commercial appeal [31,32]. Artists only hold rights to specific designs, not to creating entirely new objects [33]. Effective protection strategies are crucial for maintaining cultural integrity [1], but their application in actual product development remains limited. Bridging this gap requires integrating heritage values with contemporary design concepts, which will be further explored in subsequent sections.

### 2.2. Design principles and philosophy for ethical Souvenirs

Design philosophy refers to the values and conceptual orientations guiding a designer's creative and strategic decisions [34]. It provides the overarching value framework, while design principles translate these values into actionable guidelines. In heritage souvenir production, ethical design principles require that products meet functional and aesthetic standards while respecting cultural contexts, engaging local communities, and supporting long-term heritage stewardship [19].

Researchers identify several key principles of ethical souvenir design. Cultural respect includes protecting Indigenous intellectual property and fostering responsible engagement with local communities [35]. Sustainability involves the use of eco-friendly materials and production methods to support both environmental and cultural viability [36]. Aesthetic quality, such as symmetry, craftsmanship, and balanced design, affects consumer perception and purchase intentions [37,38]. Integrating local cultural narratives into product design strengthens authenticity and place attachment [39]. Emotional connection, fostered through designs that evoke personal or cultural memories, can further enhance the souvenir's role as a cultural bridge [40].

These design principles are rooted in cultural respect, sustainability, and community participation, forming the ethical benchmark for heritage product creation. However, applying these principles in the reality of the commercial souvenir market raises complex issues such as cultural ownership, cultural representation, and economic equity, which will be explored in the section on ethical considerations in heritage commercialization below.

## 2.3 Design ethical considerations in commercializing heritage

Ethical considerations in design extend beyond compliance with rules to encompass a reflective and context-sensitive practice rooted in authenticity and responsibility. Buwert's ethical design thinking highlights the need for designers to critically assess moral implications, treating authenticity not as a stylistic choice but as an ethical commitment that transforms potential into meaningful outcomes [41]. This aligns with Sartre's existentialist view that moral agency emerges from a conscious rejection of "bad faith," requiring designers to act with awareness of their freedom and social responsibility [42]. From this perspective, ethical design is anchored in phronesis (practical wisdom), where aesthetic sensitivity and moral discernment guide the creation of products that are not only functional and visually compelling but also socially responsible [43].

Within the souvenir design, preserving cultural integrity is both a moral and philosophical obligation. Virtue ethics positions the designer as a steward of heritage, prioritizing the safeguarding of historical, social, and cultural significance over purely economic or aesthetic gains [42]. This responsibility entails moderation and sensitivity to avoid interventions that compromise authenticity. As Cummings and Mulvenna et al. suggest, embedding ethical principles into the design process ensures alignment with societal values while promoting transparency, honesty, and inclusivity in cultural representation [44,45].

Emerging technologies in souvenir personalization introduce new opportunities but also ethical tensions, particularly in the use of Indigenous cultural symbols. Designers and commercial entities must prevent cultural misappropriation or desecration during production and distribution [46]. Yet, there remains no clear consensus on corporate social responsibility frameworks in the sensitivities of Aboriginal communities [47]. This gap underscores the need for robust, participatory approaches where communities are actively involved in decision-making to ensure respectful and mutually beneficial outcomes [48,49].

Ethical souvenir design operates at the intersection of cultural preservation, creative practice, and social accountability. By integrating moral reasoning with creative innovation, designers can ensure that souvenirs function not only as commodities but also as vehicles for cultural continuity and mutual respect.

## 2.4 Conceptual framework for ethical heritage Souvenir design

Most existing literature has explored the areas in isolation, and this separation has limited our understanding of how ethical considerations can reconcile the conflicting demands of market needs and cultural preservation. This study views commercialization as a potential driver of heritage sustainability rather than an obstacle. A conceptual framework is proposed that places cultural integrity, design principles and philosophy, and design ethical considerations as three interdependent components within the context of commercialized heritage (Fig 1). Cultural integrity ensures authenticity and meaning; design principles and philosophy serve as the foundation for creativity and strategy; and ethical considerations ensure social responsibility and long-term management. These elements are realized through design, transforming heritage value into commercially viable products.

## 3. Methodology

This study employed the qualitative case study method to examine how designers incorporated ethical design principles into souvenirs to preserve the integrity of Dunhuang cultural heritage. A two-tier purposive sampling strategy was adopted by following the principle of in-depth contextual inquiry [50,51]. Purposive sampling was chosen because the research required information-rich cases and participants with specific expertise in heritage-related design, which could not be obtained through random or convenience sampling [52].

At the first level, the creative product development sector of the Dunhuang Academy is a representative case based on four criteria: (1) institutional authorization by cultural heritage protection bodies to ensure adherence to conservation

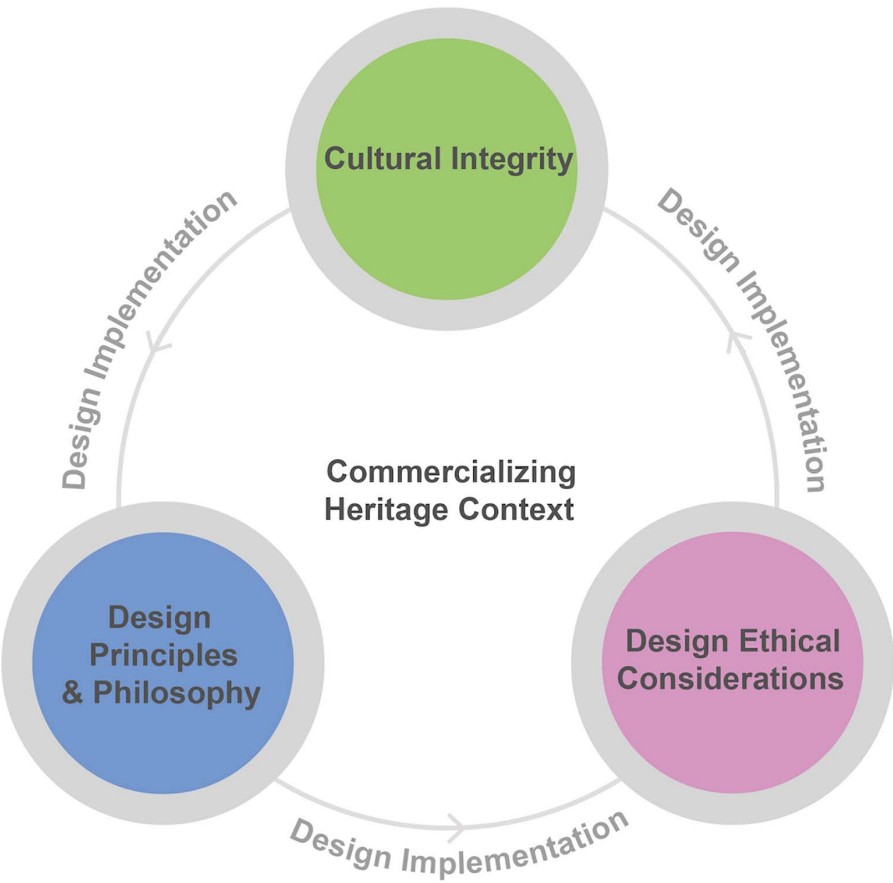

**Fig 1. Conceptual Framework for Ethical Heritage Souvenir Design.**

principles and respect for cultural integrity; (2) high relevance of products to local heritage to guarantee a close link between design outcomes and the site's cultural narratives; (3) proven capacity for creativity and innovation to produce designs that balance cultural authenticity with contemporary appeal; and (4) accessibility of design processes, personnel, and artifacts for comprehensive data collection. At the second level, participants were recruited from within the selected cases using purposive criteria: (1) professional involvement in the design, planning, or curation of heritage-themed souvenirs; (2) a minimum of one year's relevant work experience; and (3) direct engagement with Dunhuang cultural elements in their practice. This targeted approach can offer substantive, experience-based insights into the research questions [53].

### 3.1. Data collection and analysis

Interviews, observations, and document analysis are appropriate data collection methods in this investigation [54,55] that help the researcher obtain consistent conclusions and more accurate results. Participant recruitment took place from August 2022 to January 2023. Preliminary data collection was conducted in Dunhuang from September 2022 to July 2023. Additional follow-up interviews were conducted under the same ethical approval in August 2005. Interviews were the primary data collection method, targeting eight designers and three cultural experts who were selected for semi-structured interviews. After the initial analysis, two additional participants (one designer and one cultural expert) were recruited to increase the robustness of the study. To achieve the research objectives, all participants must meet the criteria for purposive sampling

described above. A total of 14 interviews were conducted, including three additional interviews with designers as a pilot study to test content validity, clarity of wording, sequencing, timing, and logistics of the interview protocol. In qualitative inquiry, small pilots (≈2–5) are standard because the purpose is protocol refinement rather than saturation or theory building [56,57]. Insights from the pilot led to minor revisions to the interview questions; pilot data were not included in the final analysis.

The researchers crafted a set of interview questions to build a theory and answer the research questions, which are semi-structured, in-depth interviews designed to provide a comprehensive view of a complex internal situation [58]. The interviews lasted approximately 30–60 minutes and were conducted in Chinese with a digital recording. Participant information is summarized in Table 1 below. Observations were also conducted at online and offline souvenir shops, design company offices, and cultural heritage sites. Additional design material from the designers was obtained to triangulate and supplement the interview data.

Data analysis and data collection were designed as an iterative process, with insights from earlier interviews informing subsequent data gathering to strengthen overall convergence [59]. The analysis adopted an inductive approach [60] to answer the research questions. Thematic analysis, recognized for its flexibility in qualitative research, was selected as it accommodates theoretical exploration in complex contexts [61]. Data management and coding were facilitated by ATLAS. ti 24, and categories emerged from clustering comparable data, allowing researchers to describe the data systematically [62]. Saturation was operationalized through two criteria: (i) code saturation, whereby no new codes emerged across two consecutive interviews; and (ii) meaning saturation, when sufficient richness and variation had been captured to delineate the properties of existing codes [63]. These criteria were satisfied after the ninth interview. To ensure stability and confirm the robustness of the findings, two additional interviews were conducted, which produced no new codes or themes [64].

### 3.2. Trustworthiness and ethical considerations

The validity and reliability of a study depend on the ethics of the researcher [65]. Before data collection, the researcher obtained approval from the University Ethics Committee. The study was conducted under the Declaration of Helsinki and approved by the Science and Technology Department of Lanzhou University of Finance and Economics (Ref No. LUFE-2022–0721). For confidentiality, pseudonyms were used instead of the participants' real names. All participants volunteered to participate in the study and signed an informed consent form, were treated with total respect during the investigation, and were allowed to check all documents concerning them. The form included (i) the purpose and procedures of

**Table 1. Detailed information of participants.**

| Category Summary | Pseudonym | Role | Gender | Age | Education Level | Experience |
|---|---|---|---|---|---|---|
| Designers (n = 8) | D1 | Design Director | M | 30 | Specialty | 7 years |
| | D2 | Design Director | F | 30 | Bachelor | 7 years |
| | D3 | Illustrator | F | 28 | Bachelor | 6 years |
| | D4 | Illustrator | M | 30 | Bachelor | 3 years |
| | D5 | Director of Planning | F | 29 | Master | 2 years |
| | D6 | Design Director | F | 26 | Bachelor | 3 years |
| | D7 | Product Designer | M | 27 | Bachelor | 1 year |
| | D8 | Product Designer | F | 28 | Bachelor | 3 years |
| Cultural Experts (n = 3) | E1 | Cultural Expert | M | 51 | Bachelor | 22 years |
| | E2 | Cultural Expert | M | 67 | Specialty | 30 years |
| | E3 | Product Development Manager | F | 41 | Master | 5 years |

Note. Eight designers (D1–D8) and three cultural experts (E1–E3) participated in the study. The table summarizes the participants' demographic and professional profiles. Pseudonyms are used for anonymity. Please note that participant age was recorded when data were collected.

the study; (ii) the voluntary nature of participation; (iii) the right to withdraw at any stage without penalty; (iv) assurance of anonymity and confidentiality; and (v) agreement for audio recording and use of data solely for academic purposes. The researcher can be alert to all possible scenarios during the data collection process, avoiding preconceived assumptions and predictions based on his or her own experience while understanding the participants' perspectives.

The researcher was free of personal biases that could influence the research topic [66]. As researchers independent of the profession, their potential biases were addressed through continuous reflexivity and qualitative-appropriate strategies. Before and throughout data collection, a reflexive journal was maintained to document assumptions, positionality, and evolving reflections [67]. In addition, a pilot interview was used to refine questions, and triangulation, member checking, trail audits, and peer feedback further helped mitigate predispositions. These practices allowed the researcher to critically examine personal assumptions and minimize their influence on data collection and interpretation. The authors also used the kappa statistic throughout the coding process to ensure data consistency. Kappa is a robust statistical method used to assess the degree of agreement between coders beyond what would be expected by chance [68], thus increasing the credibility of the study results. Two authors coded the interview transcripts separately using an inductive approach, and a confusion matrix was constructed to assess the consistency of the two themes. Out of 50 coded instances, the observed consistency ($p_o = \frac{45}{50} = 0.9$) and the expected consistency ($p_e = 0.5576$) were calculated from the distribution of codes. The calculated Cohen's kappa value $k = \frac{p_o - p_e}{1 - p_e} = 0.77$ indicated general agreement between the two authors [69,70].

Fig 2 shows the strong associations between trustworthiness factors and key measures to ensure research credibility and validity.

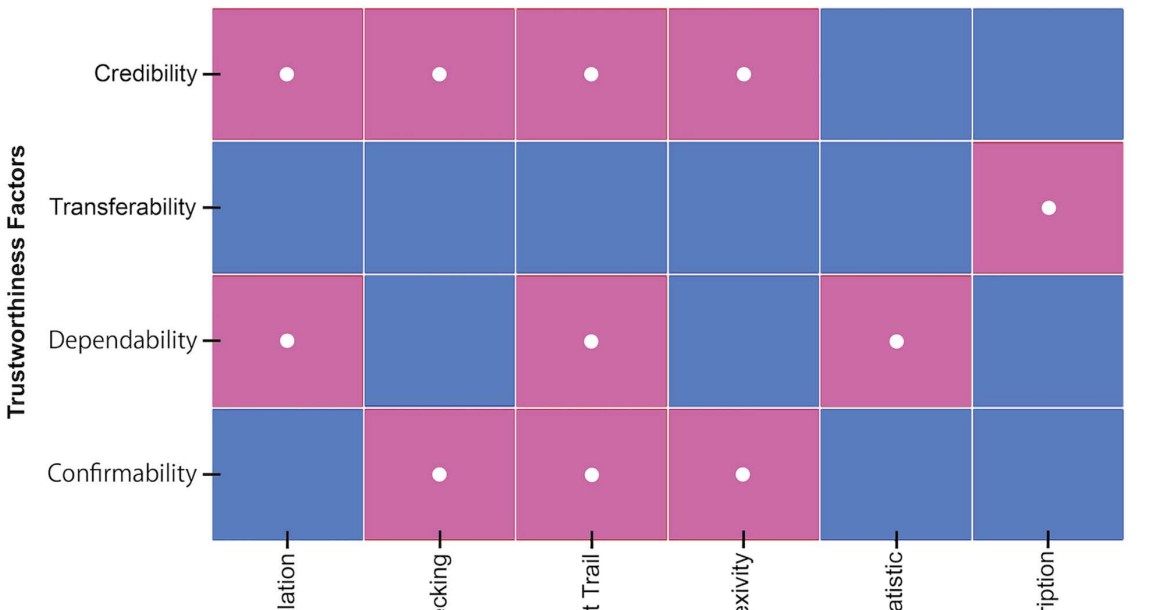

**Fig 2. Trustworthiness atrix.**

## 4. Result

A total of 59 initial codes were generated from the interview transcripts and supplementary documents (see S1 Dataset). Through iterative clustering and abstraction, these codes were consolidated into eight categories and three overarching themes: (1) Design Principles and Philosophy, (2) Design Ethical Considerations, and (3) Design Implementation. Following a reflexive thematic analysis, coding decisions were repeatedly revisited to ensure transparency and conceptual rigor. The progression from raw data to thematic abstraction is illustrated in Fig 3. Design principles and philosophy establish the cultural anchors and set out how Dunhuang motifs are translated into aesthetic and functional product terms. Design ethical considerations provide the boundary conditions and procedures—cultural sensitivity, authorization, and intellectual property—that moderate these choices and ensure legitimacy. Design implementation operationalizes the resulting decisions through prototyping, trial exposure, and revision under cost, material, and process constraints, and the evidence it generates feeds back to refine both principles and ethical rules. In combination, the three themes form an iterative system in which anchoring, governance, and execution co-produce cultural integrity and market relevance. Table 2 summarizes the final thematic framework with representative quotations.

### 4.1. Design principles and philosophy

The fundamental philosophy of a company presents the values and principles behind its creation, and this uniqueness is essential for gaining recognition and creating a solid connection with the audience. Design principles guide both the

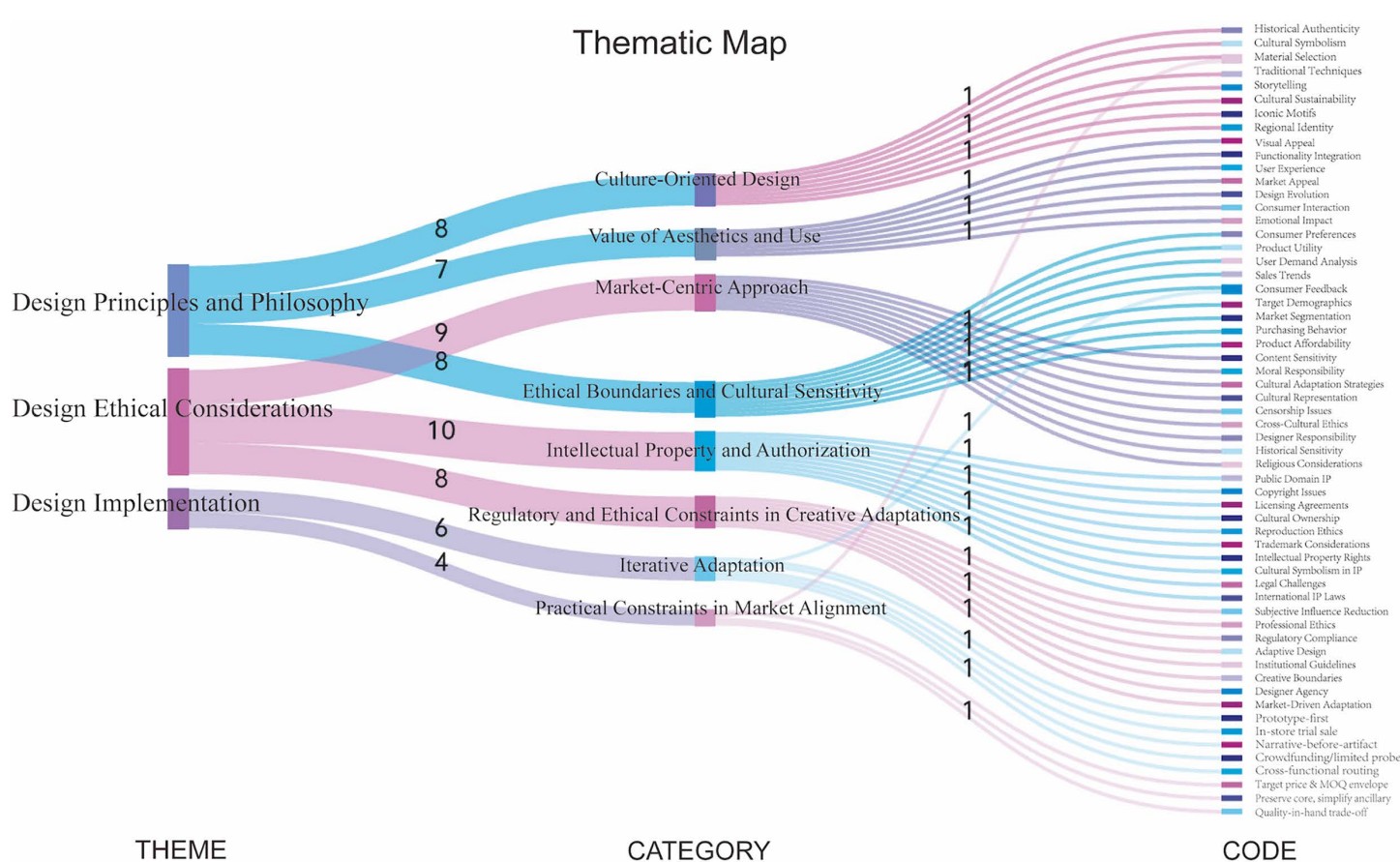

**Fig 3. Inductive analysis from codes to themes of philosophy and ethics in souvenir design.**

**Table 2. Thematic map of the designer philosophy and ethics on cultural integrity.**

| Theme | Sub-themes | Description |
|---|---|---|
| Design Principles and Philosophy | Culture-Oriented Design | "It is crucial to showcase Dunhuang's artistic splendor through tangible souvenirs, enabling the public to experience the magnificence of Mogao art." |
| | Value of Aesthetics and Use | "The initial approach involves integrating traditional murals into items, such as blending the Protective Deities from Tian-long Babu with the popular and appealing blind box designs." |
| | Market-Centric Approach | "It is crucial to plan product categories based on what customers like and what is useful." |
| Design Ethical Considerations | Ethical Boundaries and Cultural Sensitivity | "From the perspective of design and production, it is necessary to avoid some not-so-good elements or contents." |
| | Intellectual Property and Authorization | "The Mogao Cave murals are considered public intellectual property (IP), which means that their copyright has returned to the public domain." |
| | Regulatory and Ethical Constraints in Creative Adaptations | "However, as designers gradually accumulate experience and have a deeper understanding of the design content, the influence of personal subjectivity will decrease." |
| Design Implementation | Iterative Adaptation | "We usually need several rounds of prototypes and modifications before we can settle on a final form." |
| | Practical Constraints in Market Alignment | "We usually need several rounds of prototypes and modifications before we can settle on a final form." |

symbolic and practical dimensions of product creation, shaping authenticity, usability, and market relevance [71]. In the innovation of Dunhuang souvenirs, these principles represent the normative foundations through which cultural products are conceptualized and positioned. They structure the translation of cultural narratives into material form and reveal three interrelated orientations: embedding cultural identity within design, integrating aesthetic qualities with practical use, and aligning production with market expectations. These orientations illustrate how designers negotiate the tension between heritage preservation and consumer appeal, thereby linking design philosophy to the maintenance of cultural integrity.

**4.1.1. Culture-oriented design.** Culture-oriented design advocates embedding local heritage and symbolic meaning in product form, ensuring that commercial souvenirs narrate local culture. D4 notes that the critical role of cultural conveyance is "When I design, my main goal is to convey its cultural content." D2 shares this view of "the indispensability of incorporating local and artistic traits so that everyone can recognize it as Dunhuang's at first glance." In contrast, D3, affiliated with Dunhuang Academy, adheres to classical values and official norms.

> It is crucial to showcase Dunhuang's artistic splendor through tangible souvenirs, enabling the public to experience the magnificence of Mogao art. Additionally, establishing a connection between the product and Dunhuang's identity, whether through color, shape, or intrinsic story, is considered essential. (D3)

As D2 and D4 emphasize, "There is a necessity for a product to be unique and instantly recognizable as distinctly Dunhuang." Additionally, D4 states that cultural representation through a product can evoke feelings of connection and empathy, acting as a souvenir and fostering users' appreciation for the destination.

> When I visit a place, I purchase souvenirs that embody the local culture; this makes me feel empathetic. Seeing the product reminds me of the destination, so I buy it as a gift for my friend. At that moment, I believe the product is successful. (D4)

The philosophy outlined by D6 supports the idea that "cultural creative products should not be isolated artifacts but should integrate into our lives, serving the public and fulfilling cultural and societal needs...therefore making life happier." However, the overexploitation of cultural elements should be avoided during the application process. In the competitive field of souvenir design, D4 presents their firm's unique design philosophy that sets them apart from competitors. "The Dunhuang Museum's designs are trendier, featuring more exaggerated depictions, such as a Buddha smoking or skateboarding, which are unacceptable to our partner, the Dunhuang Academy." Instead, they focus on "creating more reserved and subdued products, offering a more refined and restrained expression of cultural elements." This stylistic divergence underscores the differences in interpreting and representing cultural narratives and symbols within creative products.

D2's approach emphasizes not just aesthetics but a protective sentiment towards Dunhuang. "We have been conveying the idea of protecting Dunhuang to our customers, thereby cultivating a group of Buddhist pilgrims or contemporary guardians." This spirit is reflected in their slogan, "Fighting against decay, reflecting the millennium of existence and constant changes of the Mogao Grottoes in our products." This is a conceptual stance and a call to everyone interacting with their products to "join the fight against time and make the Mogao Grottoes last longer." Her colleague D3 added, "It is important to raise public awareness of protecting cultural heritage through their souvenirs."

This symbiotic approach is characterized by a mutual interaction where the Dunhuang Cultural and Tourism Group aims to integrate Dunhuang culture across various sectors. This is evident in their design concept of "integrating Dunhuang into every one of the 360 industries" (D5). As stated by their design company, the principle of "an inch wide and a hundred feet deep" (D5) shows Dunhuang's profound and extensive influence in diverse domains. D5 proposes the specific initiatives of relating Intangible Cultural Heritage to the features of Dunhuang.

> The concept of Intangible Cultural Heritage + Dunhuang originated from the idea proposed by our chairman to integrate Dunhuang heritage into various industries. It is also a consistent core concept in the development of the Dunhuang Cultural Tourism Group. I try to apply intangible cultural heritage technology to the product when designing. The primary goal is to embed Dunhuang's rich intellectual property and craft concepts within various industries, encouraging collaborative efforts to disseminate Dunhuang culture more widely. (D5)

Furthermore, D3 argues that the "identifiability" of a company or product depends on retaining its core elements.

> As long as the core elements are maintained, the city's identity or the company's products will always be recognizable. Once you remove these foundational aspects, the product's recognition fades. This is fundamentally our bottom line; even if we reimagine or redesign, the basic elements must remain intact and untouchable. Innovation is encouraged but must be done thoughtfully and not arbitrarily. (D3)

These designers collectively advocate for products that are uniquely identifiable as Dunhuang while avoiding overexploitation and fostering a protective sentiment towards cultural heritage.

**4.1.2. Value of aesthetics and use.** Aesthetic appeal and practical functionality shape consumer perception and long-term engagement, as designers aim to merge visual attraction with everyday usability. D4 pointed out the role of aesthetic appeal in product design, acknowledging that "products with high aesthetic value will attract a certain group of customers who value appearance before understanding its connotation because the first impression is often based on the appearance of the product." The divergence of approaches between visual appeal and adherence to classics requires balancing aesthetic needs and cultural connotations, which is the key to attracting consumers and the success of products. D4 also illustrates that the rate of use in daily life largely determines the products' success. This perspective aligns with the broader vision of transforming souvenirs into everyday items, ensuring their intrinsic value and meaning are fully realized and continued. The synergy between aesthetic appeal and cultural expression captures the evolving spirit of

Dunhuang cultural creation. D1 proposes that their design philosophy merges art and utility to create meaningful designs. D4 cites a product as an example (Figs 4 and 5). She describes,

> The initial approach involves integrating traditional murals into items, such as blending the Protective Deities from Tianlong Babu with popular and appealing blind box designs. This intentional fusion of old and new utilizes contemporary color design techniques, illustrations, and various design methods to integrate cultural elements. The result is products that can either be trendy or faithful reproductions, depending on the nature and purpose of each piece. (D4)

This strategy enables the creation of various products that cater to various market needs and preferences, ensuring the continuation and preservation of traditional elements while incorporating modern trends and aesthetics. However, as a cultural expert, E2 is concerned about the product's sustainability.

> At present, we can call it cultural creation, mainly the creation of cultural commodities. Commodities have market demand. All of China is developing a less clear culture. Products may have similarities, and there may be market saturation. So, from the perspective of commodities, you can make a special one, but it is not easy. We hope to see this kind of enduring cultural product. (E2)

E2 reveals the complexity and untapped potential in the souvenir design process, suggesting that innovative approaches that tap into cultural structures are needed to meet contemporary market challenges.

**4.1.3. Market-centric approach.** Market-centric design reflects a strategy of matching product categories with consumers' desire for cultural integrity. D5 describes a distinctive strategy for product development that seeks to balance commercial appeal with cultural authenticity. He explains this approach as follows:

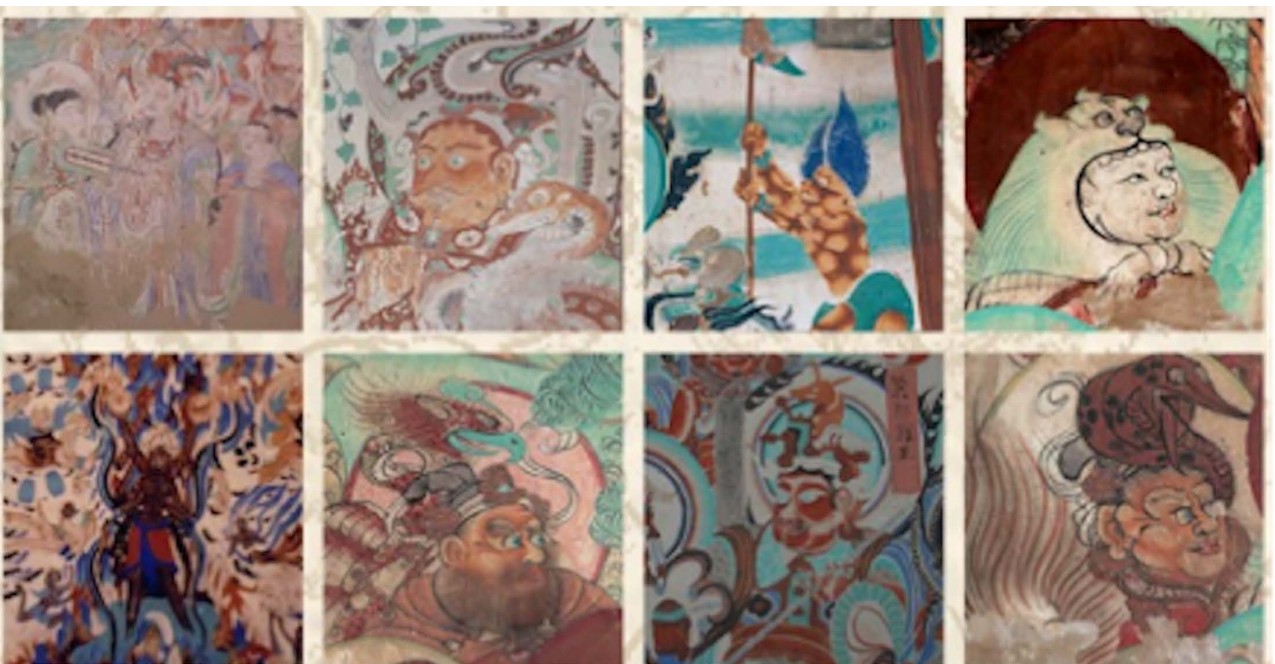

**Fig 4. The Motif of the Blind Box Design: Tianlong Babu in the Murals.**

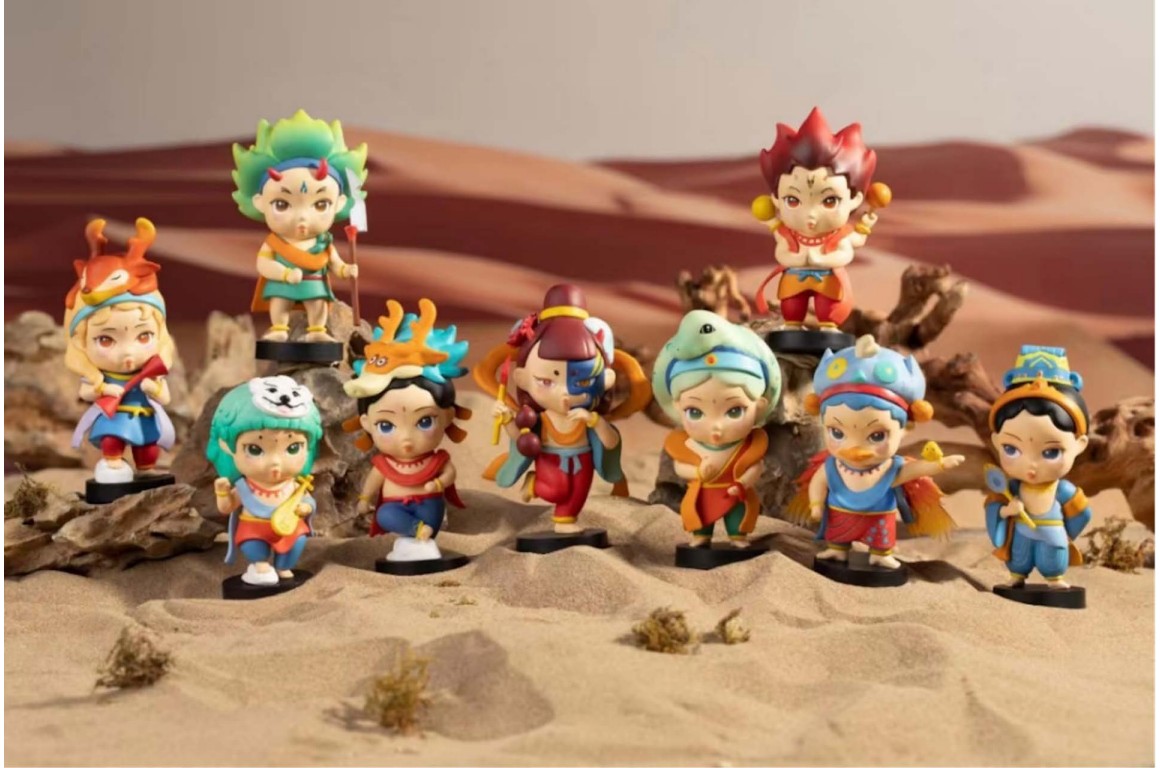

**Fig 5. Tianlong Babu Blind Box.**

> Approximately 60% of our products are general items that are affordable and practical for most people, appealing to a wide audience. Around 30% are business gifts, which may be used less frequently or are aimed at higher-income consumers. These items are typically priced under 200 yuan, or between 600 and 1000 yuan. The final 10% consists of trend-research products, which are more experimental, with costs primarily associated with extended research and development. We call it the 6-3-1 model. (D5)

D2 and D7 illustrate the necessity of combining product design and selection with consumer preferences, needs, and contemporary trends. D7 proposed a dualism of design, "On the one hand, we have to reflect Dunhuang culture, and on the other hand, we have to consider how to optimize sales…It is crucial to plan product categories based on what customers like and what is useful." D2 agreed with this view and emphasized that "products must be in harmony with current popular elements and the aesthetic preferences of the younger generation. A good example is the popularity of plush toys, which reflects current consumer trends." These insights tell the relationship between contemporary product design and selection aesthetics, cultural representations, and consumer needs. It also provides a new 6-3-1 market strategy for differentiating consumers.

### 4.2. Design ethical considerations

Design principles establish guidelines for shaping the core values of cultural products, which link to ethical responsibilities in practice. Ethical considerations serve as a moral compass, guiding the process of transforming Dunhuang heritage into authentic souvenirs. These considerations include respect for cultural boundaries, appropriate authorization of usage

rights, and compliance with broader regulatory norms. The sub-themes represent the tension between creative freedom and cultural responsibility, reminding designers to innovate products that embody cultural dignity.

**4.2.1. Ethical boundaries and cultural sensitivity.** Designer ethics and cultural sensitivity are crucial, and designs should avoid any symbolism or interpretation seen as inappropriate or disrespectful as much as possible. E2 believes that designers must maintain respect for the sanctity of cultural symbols, such as not inappropriately placing revered elements in inappropriate environments, and "painting sacred elements of temples on the soles of shoes" is an obvious example of a lack of respect in design from the real world. When a design blatantly "puts a Buddha statue on a mask, or draws a camel on it," it illustrates the problem of using cultural elements in a non-cautious design. Therefore, he advocates that "designers must know the most basic rules, they must be respectful...avoid violating major taboos", which requires "designers must be knowledgeable." Designer D3 further confirmed this view, asserting that "some basic things should never be tampered with as time goes by." D2 also supported their view and gave an example that "when using the Buddha image, it should not be given any inappropriate meaning, such as excessive malice or ironic humor." She resisted insensitive or arbitrary changes to these traditional elements and believed that products should convey positive values and emotions, consistent with the broader goal of protecting cultural heritage. The harmonious integration of respect, compliance with basic norms, and innovation became the core principles that guided the utilization of the Mogao Caves' cultural elements. D6 and D7 similarly pointed out that respecting cultural heritage is a crucial ethical consideration in the design process.

> From the perspective of design and production, it is necessary to avoid some not-so-good elements or contents… because they may have a bad impact on the cultural heritage and make it incomplete. As a result, it may cause consumers to misunderstand the cultural elements. (D7)

D7 suggested that designers should be cautious: "Designers cannot over-consume cultural heritage," and be vigilant and responsible when approaching cultural heritage, especially in the context of cultural heritage being commercialized or otherwise disseminated. D6 summarizes the potential dilemma facing designers: "Sometimes, designers overthink or overcomplicate the design process." E2 also advocates for a more ethical approach to products that "guide" rather than "allure" consumption, aligning it with critical cultural norms and values. Cultural creators are responsible for shaping social values and behaviors through their products. In summary, designers must innovate with reverence to protect the sanctity and integrity of cultural heritage while processing creative expression.

**4.2.2. Intellectual property and authorization.** Depicting and using the Dunhuang heritage requires a certain level of authority and respect. "The Mogao Cave murals are considered public intellectual property (IP), which means that their copyright has returned to the public domain" (D5). Therefore, restrictions on using elements are flexible, but authority and context in the field are crucial for credibility. D5 stated, "Organizations such as the Dunhuang Academy, Dunhuang Cultural Tourism Group, and Dunhuang Bookstore have a certain level of authority." While using the original murals may require authorization from the Dunhuang Academy, derivative works interpreted by designers do not need their approval or consent. D3 raised his concerns:

> Even a skilled top designer may not be able to truly depict the unique essence of the murals. Design is not only about harnessing expertise but also about capturing the spirit of the heritage. Therefore, it is crucial to maintain a deep understanding and reverence for Dunhuang culture and history. (D3)

**4.2.3. Regulatory and ethical constraints in creative adaptations.** Designers have a clear and careful demarcation of what is and is not allowed when adapting Dunhuang's cultural heritage elements. D3 explained the inherent limitations of reinterpreting elements of classical murals, attributing these limitations to personal principles and institutional

supervision: "First, they are reluctant to change or modernize classical elements radically, and second, they are strictly supervised by the Dunhuang Academy." Specifically, while the Dunhuang Academy is working hard to present and recreate elements of the Mogao Caves, its affiliated design companies must strictly maintain the sanctity of Dunhuang's cultural heritage in their products. "Some design strategies may damage the heritage image, such as the collaboration between the Dunhuang Museum and e-cigarette brands, which is strictly prohibited in our company" (D3). D7 affirmed this cautious approach, emphasizing the need to review products that may cause controversy carefully and may also abolish them to maintain the collective commitment to the integrity of heritage. E2 criticized the nature of some contemporary cultural and creative products: "Today, many cultural and creative products are created just for the sake of cultural creativity and are designed to attract the attention of the younger generation." Because the superficial use of cultural elements may weaken their intrinsic value. He also reflected on such products from the perspective of human ecology, believing that "they are unfriendly and deviate from the principles of conservation and protection emphasized in traditional values." This view expresses the need for more ethical and sustainable considerations when creating and consuming cultural products.

In addition, designers may have personal biases at the beginning of design. "However, as designers gradually accumulate experience and have a deeper understanding of the design content, the influence of personal subjectivity will decrease." (D7) When reflecting on the impact of design concepts, D5 admitted that their design methods have a hidden effect on consumers. D6 also realized the challenges brought by the pioneering ideas of some designers:

> The main problem may be that some designers' ideas are too advanced. This creates a problem when our products are mainly presented to the general customer base, and they may not understand the design concept immediately. Therefore, the product is unacceptable. (D6)

In summary, designers believe that creative freedom should be promoted within the scope of maintaining reverence and respect for cultural elements, and social norms should be adhered to when using cultural symbols to preserve the lasting dignity and integrity of Dunhuang's cultural heritage.

### 4.3. Design implementation

This theme shows how the philosophical and ethical orientations identified earlier are translated into concrete practice. Implementation unfolds through continuous iteration and through decisions shaped by materials, processes, cost, and market fit.

**4.3.1. Iterative adaptation.** Designers consistently described the development as cyclical rather than linear. D3, D5, and D8 point out that a product rarely succeeds on the first attempt: "We usually need several rounds of prototypes and modifications before we can settle on a final form." D3 explained that "craft and process concerns are already on the table at the earliest stage of design, not left until the end." D8 further notes that after initial prototyping, teams often conduct small-scale trials in retail spaces to see how customers interact with products. "Placing limited runs in the store is the simplest way to judge if the design deserves a wider release (D5)." Feedback then drives targeted changes. Another interviewee (D4) noted, "I often follow up with buyers in shops, collect their opinions, and send those directly to our planners and production staff so the design can be adjusted."

Designers also leverage online channels. They post cultural background stories on public platforms and provide iconographic explanations on product pages. Yet, if sales remain unsatisfactory, they will revise motifs and experiment with alternatives. As one observed (D6), "When a theme doesn't resonate, we switch to another cultural element that customers can more easily connect with." Rapid shifts in consumer taste, such as the blind-box craze, demand responsiveness, and crowdfunding, are widely used as a safe testing ground (D3, D7). Overall, iteration serves as a forward-looking mechanism that integrates cultural meaning with market feedback.

 

**4.3.2. Practical constraints in market alignment.** Implementation also depends on decisions that balance authenticity with manufacturability. For replica-based products, interviewees emphasized the use of faithful materials and techniques to retain historical texture. At the same time, they adapted details for modern production. One designer (D2) explained, "We designed the porcelain with engraved patterns on the back, using steel molds and additional printing to create a tactile quality that feels premium and long-lasting." Similarly, glass bookmarks blending intangible craft and contemporary production have proven commercially viable; since 2022, one line has sold nearly a thousand units (E3). Teams also institutionalize quality control. For example, when producing a Dunhuang calendar, D5 described how "all texts and images were prepared early, and professional reviewers from the publisher checked and proofed the content before anything was printed."

D4 identified the inherent conflict in product development between maintaining artistic values and complying with market constraints:

> During the implementation phase, factors such as price and minimum order quantity may limit our ability to improve the product, which may require reducing some design content. But we do not want to make a profit by selling the product at a low price at the beginning. (D4)

This approach ensures that the pursuit of profit does not compromise the essence of design. Cost, material choice, and production methods determine what can be realized at scale, while fidelity to cultural meaning sets boundaries for adaptation.

Design implementation is the practical process of validating the feasibility of design principles and philosophical frameworks within real-world constraints. It guides designers to respect cultural boundaries, transform Dunhuang heritage into tangible souvenirs, secure authorization of usage rights, and comply with regulatory norms to prevent misuse. In this way, the implementation phase bridges the concept-driven innovation with ethical responsibility, which reflects an iterative process of creative intent, review, and market adaptation. This interconnection demonstrates that cultural authenticity and consumer appeal can be maintained when philosophy, ethics, and practice operate in tandem.

## 4.4. A framework of ethical souvenir design for cultural integrity

The output of this study is a Framework of Ethical Design for Cultural Integrity that integrates design principles and ethical considerations to protect cultural integrity in souvenir making (see Fig 6). It emphasizes culturally oriented design, aesthetics, and market relevance while addressing ethical obligations such as cultural sensitivity, intellectual property rights, and compliance. It identifies design principles that combine tradition and modernity in the souvenir innovation process and proposes a responsible approach to innovative product design by balancing commercial and cultural interests to achieve the goal of cultural integrity as defined by UNESCO. However, the framework is based on the design practices in the case study and has not yet been validated in practice. In the future, it could be applied to other cultural product development to test the usability. Meanwhile, the framework combines cultural integrity and design ethics to construct a theory of innovative design centered on cultural values preservation.

## 5. Discussion

Our analysis of souvenir design focuses on the interplay between cultural preservation and market-driven innovation. Designers face the challenges of merging tradition and modernity, balancing commercial and cultural interests, and assuming evolving responsibilities within ethically sensitive frameworks. They emphasize authentic cultural narratives and favor unique designs that resonate emotionally with consumers. They prioritize quality over quantity and maintain cultural integrity, setting strict boundaries when adapting creative elements. These findings respond to the goal of this study, which is the critical role of design concepts and ethical considerations in addressing these challenges.

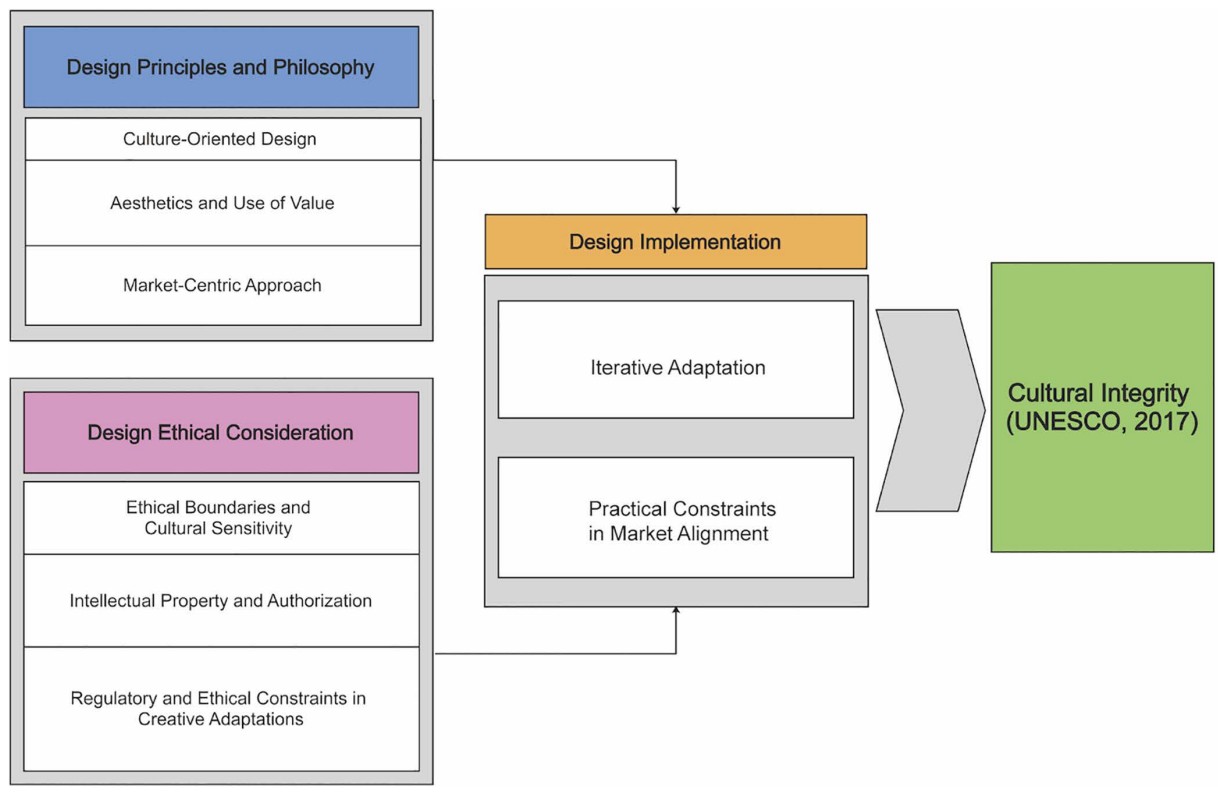

**Fig 6. Framework of Ethical Design for Cultural Integrity.**

## 5.1. Design principles and philosophy for preserving cultural integrity

Addressing RQ1 and RQ2, this study examines how design philosophy and ethical principles guide the translation of Dunhuang mural motifs into products that maintain cultural integrity while remaining intelligible and usable in contemporary markets. The findings indicate that designers treat cultural integrity as a core benchmark. They preserve recognisable cultural anchors to foster emotional connection and coordinate colour, material, and form so that aesthetic value aligns with everyday utility. These practices embody key ethical commitments, including respect for sacred and symbolic elements, a sustained concern for narrative authenticity, and responsibility for audience understanding.

These conclusions are consistent with and extend prior research. Work on culturally sensitive design places authenticity at the centre of product development. Our data show that authenticity is realised through systematic choices of symbols, materials, and narrative presentation rather than through abstract declarations [72]. Studies of cultural and creative products identify affect as a driver of preference. The present analysis adds that emotional connection, local narrative, tactile craft qualities, and usability operate in concert to sustain user engagement in everyday settings [73]. At the level of process, scholarship on design reasoning emphasises innovation through problem framing and reframing. Designers in this study did not make a one-time choice between tradition and modernity; instead, they iteratively reframed briefs to convert open-ended challenges into actionable tasks, aligning with accounts of designerly reasoning and reflective practice [74,75]. Broader trends in ethical consumption and the valuation of cultural authenticity further support these pathways and help explain why perceived cultural sincerity is associated with positive evaluations and purchase intentions among visitors and consumers of heritage products [76].

Taken together, the evidence suggests that three orientations operate in concert rather than in isolation. Culture-oriented design secures recognisability and meaning. The coordination of aesthetic and use value enables everyday embedding. Audience and channel segmentation sustains market relevance without diluting cultural content. Within this configuration, design principles and philosophy serve as the conceptual substrate that steers concrete choices about motifs, materials, and narrative form.

### 5.2. Design ethics as boundary conditions for innovation

Field evidence shows that ethics is not a post-hoc remedy but an integral part of the production workflow. Designers articulated shared commitments to honour sacred and symbolic elements and to avoid uses that might demean or distort meaning. They reported routinely disclosing provenance and providing explanatory notes for the public, while maintaining clear rules that distinguish what may be adapted from what must remain intact. Where canonical imagery was involved, work proceeded under licensing agreements and institutional review. Teams also screened collaborations with sensitive industries and reduced individual subjectivity through peer review and accumulated professional experience. These arrangements preserved cultural integrity while leaving room for creative interpretation.

This configuration accords with accounts that treat design as an ethical practice grounded in respect, honesty, and responsibility in representation [42,77]. It also aligns with scholarship on Indigenous and community cultural rights, which calls for informed engagement and appropriate permission when drawing on protected symbols and traditional cultural expressions [35,78]. Recent discussions of norm-sensitive and value-aware design emphasize embedding ethics as procedures rather than relying on general statements; the observed ex-ante checks, provenance explanation, and boundary rules provide empirical support for this procedural turn [79]. In this setting, authorization and institutional oversight did not suppress creativity; they supplied the legitimacy conditions under which cultural adaptation can proceed without eroding trust or dignity.

In relation to RQ1 and RQ2, the practices specify the ethical principles of cultural sensitivity, authenticity, permission, and accountability, which guide designers towards authentic outcomes. Ethics operates here as a governance boundary that moderates the connection between adaptive creativity and cultural authenticity.

### 5.3. Design implementation within ethical constraints

In our cases, implementation functions as a bridge that converts creative intent and ethical guardrails into market outcomes through learning under real constraints. Teams move from prototyping to small-scale pilots, then to revision and selective scaling. The process is cumulative in that each stage informs the next, and no single step is treated as an endpoint.

First, designers learn from early user contact. Prototypes are handled, discussed, and observed in physical stores; small batches are placed as trial items, and short-cycle feedback is gathered through follow-ups with purchasers. Online channels add another layer of evidence through narrative posts that articulate provenance and symbolism before altering the artefact itself. Crowdfunding and limited releases operate as low-risk probes that reveal the fit between products and audience expectations. The pattern is consistent with research linking rapid experimentation and staged commitments to improved product–market fit [80].

Second, practical limits shape and often refine solutions. Target prices, minimum order quantities, material availability, and process capability require designers to streamline forms, select feasible techniques, and calibrate detail. Core motifs are preserved intact while ancillary elements are simplified or adjusted. Traditional processes and faithful materials are used when replication is the aim; hybrid combinations are adopted when manufacturability is decisive. This accords with accounts that treat constraints as resources for problem framing and iterative improvement [81].

Third, implementation embeds ethics within production rather than appending it at the end. Work that involves canonical imagery proceeds with licensing and institutional review. Teams disclose provenance and provide explanatory notes

to the public. Collaborations with sensitive sectors are screened, and peer review mitigates individual bias, which supplies legitimacy conditions for adaptation, protecting cultural dignity while leaving scope for creative interpretation [82].

Taken together, implementation advances the commitments set by design principles and ethical considerations through cycles of testing, revision, and selective expansion. Cultural anchoring is maintained by examining recognizability and narrative resonance in use. Legitimacy is sustained through authorization and provenance practices. Market relevance is preserved by aligning audience and channel choices with feasible materials and processes. With respect to RQ2, implementation operates as the mechanism that translates philosophical orientation and ethical commitments into acceptable products under realistic constraints.

## 6. Conclusion

This study employs a single-case qualitative inquiry to examine how design philosophy and ethical principles guide the translation of Dunhuang mural motifs into products that preserve cultural integrity while remaining usable in contemporary markets. The reflective thematic analysis integrates three domains into a single process view: design principles and philosophy, design ethical considerations, and design implementation within ethical constraints. The findings address RQ1 by specifying the ethical values that orient authentic outcomes, and address RQ2 by showing how philosophy and ethics, through iterative implementation, shape decisions about motifs, materials, narrative presentation, and market pathways.

### 6.1. Theoretical contributions

This study proposes a theoretical framework that integrates design concepts, ethical safeguards, and implementation processes within the commercialization of cultural heritage. It shows that culture-oriented conception, the coordination of aesthetic and use value, and market orientation operate jointly rather than in isolation to sustain cultural integrity and market relevance. Ethics is reframed from declarative statements to procedural governance, in which boundary rules, provenance practices, and authorisation routines organise creative decisions and moderate the relation between adaptive creativity and cultural authenticity. Implementation is positioned as the bridge from concept to market, where constraints such as target price bands, minimum order quantities, material availability, and process capability function as resources for problem framing and iterative improvement rather than as limitations.

### 6.2. Practical implications

The practical implications of this study are as follows. First, at the level of design principles, protect cultural anchoring through a brief-stage checklist of non-negotiable elements, and test colour, material, and form in realistic use so that craft tactility aligns with everyday function; sustain relevance through segmented product portfolios and small-batch pilots, monitored by operability, repeat use, and narrative recall. Second, at the ethical level, replace declarative commitments with embedded procedures, including ex-ante checks, a concise negative-context list, provenance and attribution routines, and auditable permissions; apply internal review and partner screening when high-salience imagery is involved. Third, at the implementation level, run a feedback-rich loop of prototyping, trial use, follow-up, and targeted redesign, and set feasible cost and minimum-order envelopes early; preserve core elements while simplifying ancillary features, and combine traditional techniques with manufacturable processes.

### 6.3. Limitations and Future Research

This study also has certain limitations. It examined only the mature souvenir market and a specific industry subset in a particular region of China, so its conclusions may not be easily generalized to other cultural heritage sites or cultural product sectors. Research evidence relies on qualitative interviews; adopting a mixed-methods approach incorporating

questionnaires or experiments could validate and expand the theoretical model. The analysis adopts a designer-centric perspective; subsequent research could incorporate consumer, policymaker, and industry viewpoints to assess how authenticity perception mechanisms and governance systems shape ethical practices.

## Supporting information

**S1 Dataset.** Thematic Analysis Codebook and RQ–Theme Mapping. This document contains all 59 codes derived from the thematic analysis, grouped into 8 categories and 3 themes with their descriptions, collectively addressing the two research questions.
(DOCX)

## Acknowledgments

We sincerely thank all participants and collaborators who contributed their time, expertise, and support to this study.

## Author contributions

**Conceptualization:** Qiuxia Zhu, Rizal Rahman.

**Data curation:** Rizal Rahman.

**Formal analysis:** Qiuxia Zhu.

**Investigation:** Qiuxia Zhu.

**Methodology:** Qiuxia Zhu.

**Supervision:** Rizal Rahman.

**Validation:** Qiuxia Zhu, Rizal Rahman.

**Visualization:** Qiuxia Zhu.

**Writing – original draft:** Qiuxia Zhu.

**Writing – review & editing:** Qiuxia Zhu.

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
