## [Decision Letter · Decision Letter 0]

12 Aug 2025

Dear Dr. Zhu,

Thank you for submitting your manuscript to PLOS ONE. After careful consideration, we feel that it has merit but does not fully meet PLOS ONE’s publication criteria as it currently stands. Therefore, we invite you to submit a revised version of the manuscript that addresses the points raised during the review process.

**ACADEMIC EDITOR: **

Based on the reviewers' feedback, please make careful revisions to the following points:

Issues regarding the timeliness of references, the structural organization of the paper, and the depth of analysis.Lack of consideration for the commercialization issues in the design of tourist souvenirs.Whether the sample size is sufficient to meet the research needs—currently, the study includes in-depth interviews with 7 souvenir designers and 2 cultural experts.

We look forward to receiving your revised manuscript.

Kind regards,

Hai-Tao Yu, ph.D.

Academic Editor

PLOS ONE

Journal Requirements:

4. Please remove all personal information, ensure that the data shared are in accordance with participant consent, and re-upload a fully anonymized data set.

Additional guidance on preparing raw data for publication can be found in our Data Policy (https://journals.plos.org/plosone/s/data-availability#loc-human-research-participant-data-and-other-sensitive-data) and in the following article: http://www.bmj.com/content/340/bmj.c181.long .

Reviewers' comments:

Reviewer's Responses to Questions

**Comments to the Author**

1. Is the manuscript technically sound, and do the data support the conclusions?

Reviewer #1: Partly

Reviewer #2: Partly

2. Has the statistical analysis been performed appropriately and rigorously?

Reviewer #1: N/A

Reviewer #2: Yes

3. Have the authors made all data underlying the findings in their manuscript fully available?

Reviewer #1: No

Reviewer #2: Yes

4. Is the manuscript presented in an intelligible fashion and written in standard English?

Reviewer #1: No

Reviewer #2: Yes

Reviewer #1: This article necessitates significant amendments prior to publication; many of the citations are outdated, and considerable changes are required in the article's structure.

Based on these extensive and substantive comments, the article appears to require significant revisions to meet the standards for publication. The numerous issues—including outdated citations, structural organization, methodological clarity, and depth of analysis—suggest that it is currently not suitable for acceptance in its present form.

Therefore, the appropriate decision would be reject, with the recommendation that the authors thoroughly revise and resubmit after addressing all the outlined concerns.

Reviewer #2: The manuscript explores and discusses the creation and sustainability of economic value by adapting cultural values to tourist products. The manuscript, which includes a qualitative research approach, included interviews with designers and cultural experts. The manuscript, which aligns its purpose and methodology, identified the following points:

- The data obtained from the research was discussed primarily focused on commercialization. Ethical issues in tourist product development regarding the sustainability of cultural values were not sufficiently emphasized. The contradictions and conflicts between the views of designers and cultural experts were not sufficiently highlighted. The issue of the degradation of cultural values for the purpose of obtaining economic benefits from tourist products was not adequately discussed. Furthermore, recent research is insufficient in the references. Outdated research was used. It would be beneficial to discuss the mentioned issues by referencing recent studies. It is recommended to discuss the negative aspects of the commercialization of cultural values.

Good luck.

**Do you want your identity to be public for this peer review?** For information about this choice, including consent withdrawal, please see our Privacy Policy

Reviewer #1: **Yes: ** Abdallah Amro

Reviewer #2: No

---

## [Author Response · Author response to Decision Letter 1]

8 Sep 2025

Dear Dr. Hai-Tao Yu and reviewers,

On behalf of my co-authors, I am pleased to resubmit our revised manuscript entitled “Designing Ethical Souvenirs to Sustain the Cultural Integrity of Dunhuang Heritage” for consideration in PLOS ONE.

We are very grateful to you and the reviewers for the constructive and detailed feedback on our previous submission. We have carefully revised the manuscript to address all concerns. The key improvements include:

• Introduction: Strengthened the emphasis on the importance of ethical design practices for cultural preservation and provided a clearer articulation of the research gap. Updated and expanded the citations to reflect the most recent scholarship.

• Literature Review: Reorganized the section to establish clearer connections between cultural integrity, design philosophy, and ethical considerations. Replaced outdated references with current studies.

• Methodology: Added a sample size section with justification for purposive sampling, elaboration on theoretical saturation, and explanation of the pilot test. Detailed the ethics approval reference, contents of the informed consent form, and researcher reflexivity.

• Results: Reorganized the sequence of theme development and corresponding figures/tables. Clarified how interview data informed all three themes and explained the linkages between design philosophy, ethical considerations, and implementation.

• Discussion: Expanded to provide deeper interpretation of the Ethical Design for Cultural Integrity (EDCI) framework and connected findings to recent research.

• Conclusion: Added explicit theoretical, practical, and methodological contributions, including actionable recommendations for ethical souvenir design practice.

We believe these revisions have significantly improved the manuscript’s clarity, rigor, and contribution, directly addressing the reviewers’ concerns. We respectfully resubmit this revised version for your kind consideration, and we sincerely hope it now meets the standards for publication in PLOS ONE.

Thank you again for the opportunity to revise and resubmit.

For details, see the Response to Reviewers letter.

Zhu Qiuxia

---

## [Decision Letter · Decision Letter 1]

25 Nov 2025

Designing ethical souvenirs to sustain the cultural integrity of Dunhuang heritage

PONE-D-25-23388R1

Dear Dr. Zhu,

We’re pleased to inform you that your manuscript has been judged scientifically suitable for publication and will be formally accepted for publication once it meets all outstanding technical requirements.

Kind regards,

Hai-Tao Yu, ph.D.

Academic Editor

PLOS ONE

Additional Editor Comments (optional):

Reviewers' comments:

Reviewer's Responses to Questions

**Comments to the Author**

Reviewer #1: All comments have been addressed

Reviewer #3: All comments have been addressed

2. Is the manuscript technically sound, and do the data support the conclusions?

Reviewer #1: Yes

Reviewer #3: Yes

3. Has the statistical analysis been performed appropriately and rigorously?

Reviewer #1: Yes

Reviewer #3: Yes

4. Have the authors made all data underlying the findings in their manuscript fully available?

Reviewer #1: Yes

Reviewer #3: Yes

5. Is the manuscript presented in an intelligible fashion and written in standard English?

Reviewer #1: Yes

Reviewer #3: Yes

Reviewer #1: Your revision has substantially improved the manuscript. All major comments from the earlier review have been fully or partially addressed:

Introduction: Now emphasizes ethical design and cultural integrity, with a well-articulated research gap and updated references.

Literature Review: Thoroughly restructured with updated citations (2018–2025) and a clear conceptual chain linking cultural integrity, design principles, and ethics, culminating in a coherent framework.

Methodology: Expanded significantly, with clear justification for purposive sampling, transparent explanation of sample size and saturation, pilot study rationale, ethics approval details, and reflexivity strategies to manage bias. The addition of coding reliability (Cohen’s kappa) and a trustworthiness matrix strengthens methodological rigor.

Results: Correctly reorganized, with theme development upfront. A new theme (Design Implementation) was developed, and connections between all three themes (principles, ethics, implementation) are now explicit. Subsections are introduced with richer framing and literature support.

Discussion: Strongly tied to the framework, structured around its three components, and well-supported by recent literature. The section now extends findings into broader scholarly conversations.

Conclusion: Theoretical and practical contributions are clearly articulated. Practical recommendations are well-grounded in the categories and codes. Limitations and future research are outlined, which further strengthens the manuscript. The only area where improvement is still needed is the explicit articulation of methodological contributions—currently implied through reflexive thematic analysis and coder reliability checks, but not clearly labeled as a contribution in its own right.

Overall, the manuscript is now much clearer, better structured, and significantly more rigorous.

Reviewer #3: The manuscript is strong and has significant potential, but would benefit from moderate to major revisions focusing on clarity, conciseness, and alignment of findings with theoretical arguments.

Addressing the issues noted above will substantially improve the manuscript’s scholarly contribution and readability.

**Do you want your identity to be public for this peer review?** For information about this choice, including consent withdrawal, please see our Privacy Policy

Reviewer #1: **Yes: ** Abdallah Amro

Reviewer #3: **Yes: ** Wanamina Bostan Ali

---

## [Editor Report · Acceptance letter]

PONE-D-25-23388R1

PLOS ONE

Dear Dr. Zhu,

I'm pleased to inform you that your manuscript has been deemed suitable for publication in PLOS ONE. Congratulations! Your manuscript is now being handed over to our production team.

Kind regards,

on behalf of

Dr. Hai-Tao Yu

Academic Editor

PLOS ONE